# Undernutrition combined with dietary mineral oil hastens depuration of stored dioxin and polychlorinated biphenyls in ewes. 1. Kinetics in blood, adipose tissue and faeces

Lucille Rey-Cadilhac[1,2], Ronan Cariou[3], Anne Ferlay[2], Catherine Jondreville[1], Carole Delavaud[2], Yannick Faulconnier[2], Sébastien Alcouffe[4], Pascal Faure[4], Philippe Marchand[3], Bruno Le Bizec[3], Stefan Jurjanz[1], Sylvain Lerch[1,5]*

1 UR AFPA, Université de Lorraine, INRAE, Nancy, France, 2 UMR Herbivores, Université Clermont Auvergne, INRAE, VetAgro Sup, Saint-Genès-Champanelle, France, 3 LABERCA, Oniris, INRAE, Nantes, France, 4 UE Herbipôle, INRAE, Saint-Genès-Champanelle, France, 5 Ruminant Research Unit, Agroscope, Posieux, Switzerland

* sylvain.lerch@agroscope.admin.ch

**Data Availability Statement:** Detailed individual data are available in the data.inra.fr repository

## Abstract

Food safety crises involving persistent organic pollutants [POPs, e.g. dioxins, polychlorinated biphenyls (PCBs), organochlorine pesticides] lead to systematic slaughter of livestock to prevent their entry into the food chain. Therefore, there is a need to develop strategies to depurate livestock moderately contaminated with POPs in order to reduce such economic and social damages. This study aimed to test a POPs depuration strategy based on undernutrition (37% of energy requirements) combined with mineral oil (10% in total dry matter intake) in nine non-lactating ewes contaminated with 2,3,7,8-tetrachlorodibenzo-*p*-dioxin (TCDD) and PCBs 126 and 153. In order to better understand the underlying mechanisms of the depuration process, POPs kinetics and body lipids dynamics were followed concomitantly over 57-day of depuration in POPs storage (adipose tissue, AT), central distribution (blood) and excretion (faeces) compartments. Faecal POPs concentrations in underfed and mineral oil supplemented ewes increased by 2.0 to 2.6-fold, but not proportionally to lipids concentration which increased by 6-fold, compared to the control ewes. Nonetheless, after 57 days of depuration in undernutrition and mineral oil supplementation, AT POPs concentrations were 1.5 to 1.6-fold higher while serum concentrations remained unchanged compared to the control ewes. This was concomitant with a decrease by 2.7-fold of the AT estimated lipids weight along the depuration period. This reduction of the volume of the storage compartment combined with the increase of POPs faecal excretion in underfed and mineral oil supplemented ewes led to a reduction by 1.5-fold of the PCB 126 AT burden, while no changes were observed for TCDD and PCB 153 burdens (vs. no change for PCB 126 and increases for TCDD and PCB 153 AT burdens in control ewes). The original approach of this study combining the fine description at once of POPs kinetic and of body lipids dynamic improved our understanding of POPs fate in the ruminant.

(open access at https://doi.org/10.15454/Z6UML7).

**Funding:** This work was supported by the "Conseil Régional de Lorraine" (Nancy, France) under Grant "Université/EPST-Région 2014" and by the "Centre National Interprofessionnel de l'Economie Laitière" (CNIEL, Paris, France) to SL. The funders had no role in study design, data collection and analysis, decision to publish, or preparation of the manuscript.

**Competing interests:** The authors have declared that no competing interests exist.

## Introduction

Persistent Organic Pollutants (POPs) encompass several molecules defined in the Stockholm Convention list such as polychlorinated biphenyls (PCBs), organochlorine pesticides (e.g. hexachlorobenzene and mirex) and dioxins (e.g. 2,3,7,8-tetrachlorodibenzo-*p*-dioxin or TCDD). All POPs have several common properties, they are toxic to animals and humans, persistent in the environment and they bioaccumulate in animal tissues [1]. Particularly, POPs bioaccumulate in fat-rich tissues and organs of animals and are transferred to fat-rich excreta. Therefore, the consumption of fat-rich food products of animal origin (fish, meat, milk and eggs) is the main route of human exposure to several POPs [2]. To face and manage such food safety issue, maximal regulatory levels for POPs in food of animal origin have been set in several countries (e.g. EC regulation n˚1259/2011). Nevertheless, several food safety crises, involving accidental contaminations of livestock by POPs, occurred in past decades [3]. As POPs depuration process is extremely slow, the contaminated livestock animals and their food products are often disposed rather than saved. Therefore, there is a real need to develop rearing strategies to depurate contaminated animals in order to reduce the deleterious social and economic damages of food contamination crisis involving POPs.

Due to their lipophilic nature, POPs behaviour within animal organism is linked to lipid dynamics, especially at the absorption, tissue distribution and excretion steps. Thus, in order to depurate livestock animals from POPs, two combined steps should be considered: i) the release of POPs from their storage compartment: i.e. the adipose tissue (AT) toward the blood, and ii) the transfer of POPs from the blood toward a non-edible excretion compartment (i.e. the faecal lipids). In ewes, body fat mobilization induced by undernutrition efficiently hastened the release of PCBs from AT toward the blood [4]. Besides, supplementing growing lambs or dairy goats diets with non-absorbable lipids enhanced hexachlorobenzene and mirex faecal output by stimulating non-biliary excretion from blood toward the digestive contents [5, 6]. In addition, the effect of non-absorbable lipid supplementation on hexachlorobenzene faecal excretion in mice was improved by 1.7-fold when combined with undernutrition [7]. Similarly, reduction of PCBs body burden in growing chickens was improved by 2.3-fold with undernutrition combined with mineral oil (MO) supplementation compared to MO supplementation alone [8]. However, the synergetic effect of both undernutrition and dietary non-absorbable lipid supplementation in ruminants remains unknown and deserves further studies. Indeed, differential responses should be expected due to the differential body size, physiology, and lipid nutrition and metabolism in ruminants when compared to rodents and birds [9].

The present study aims at assessing the efficiency of undernutrition combined with non-absorbable lipids (i.e. MO) supplementation as a strategy to decontaminate ewes from stored 2,3,7,8-tetrachlorodibenzo-*p*-dioxin, and PCBs 126 and 153. These three molecules were chosen as representatives of highly persistent POPs, characterized by a poor metabolisation and a low depuration rate in milk and meat of ruminants [10–12]. This first companion paper focuses on the kinetics of POPs concentrations in the main storage (AT), central distribution (blood) and excretion (faeces) compartments. Novel aspects include the fine description at once of POPs toxicokinetics and body lipid dynamics [i.e. characterized by indicators of body lipid mobilization: body condition score (BCS), adipocyte size changes, and plasma non-esterified fatty acid (NEFA) and beta-hydroxybutyrate (BOH) concentrations] which constitutes a key stage to better understand the influence of the dynamics of lipids on the POPs fluxes.

## Material and methods

### Ethic statement

All experimental procedures were approved by the French Ministry of Research and Higher Education (agreement n° 2357–20151008171318) after an ethic evaluation by the committee C2EA-02 (Clermont-Ferrand, France). The number of animal per group was determined according to an *a priori* experimental power test performed before the initiation of the experiment. An alpha risk of 5% and a beta risk of 80% were retained, whereas the expected differences in adipose tissue POPs concentrations between the two groups of ewes at the end of experiment, together with the expected intra-group variability, were fixed based on previous experiments and literature on ewes, goats and lambs [4–6]. *A posteriori*, significant differences ($P \leq 0.05$) between treatments for tissues POPs were observed, which confirms that the number of animals used was sufficient and properly estimated *a priori*.

### Animals and diets

Nine non-lactating and non-pregnant Romane ewes [*Ovis aries*, 5.2±1.0 years old, 64±7 kg body weight (BW) and 3.9 ±0.2 points BCS, mean±SD] from experimental unit Herbipôle (INRA UE 1414, France) were individually housed at a temperature of 15°C and under the natural light-dark cycle from January to May 2016. After 21 days of acclimation, the 93-day experiment was divided into three successive periods: i) a 27-day exposure period (days -35 to -9 of experiment with day 0 the beginning of the depuration period), during which ewes ingested POPs through a spiked concentrate incorporated to a diet covering 82% of maintenance energy requirements (MER; [13]); ii) a 8-day buffering period (days -8 to -1) during which ewes were fed with a "clean" diet (with non-spiked concentrate) covering 96% of MER; and iii) a 58-day depuration period (days 0 to +57) during which ewes were divided into two groups according to age, BW and BCS, and then assigned to one of two treatments. Four ewes received a control well-fed (96% of MER) and non-supplemented treatment (CTL), while the five other ewes received an underfed (37% of MER) and MO supplemented [10% in total dry matter (DM) intake] treatment (UFMO). Detail of the composition of the diets provided to ewes during exposure, buffering and depuration periods are presented in Table 1, whereas description of the contaminated rapeseed oil preparation and mineral oil nature are provided in S1 File.

The UFMO ewes were submitted to a 3-day dietary shift between the buffering and the depuration periods (days 0 to +2). During the whole experiment, feedstuffs were individually distributed as a total mixed ration twice daily in equal amounts at 0830 h and 1530 h. The

**Table 1. Composition of the diets (percentage of the diet dry matter)[1].**

| Item | Exposure period | Buffering period | Depuration period | |
|---|---|---|---|---|
| | CTL, UFMO | CTL, UFMO | CTL | UFMO |
| Straw | 30 | 30 | 30 | 45 |
| Hay | 30 | 30 | 30 | 45 |
| Contaminated concentrate[2] | 30 | 0 | 0 | 0 |
| Non contaminated concentrate[2] | 10 | 40 | 40 | 0 |
| Mineral oil | 0 | 0 | 0 | 10 |

[1]A control group of four ewes were well-fed and non-supplemented with mineral oil (CTL), while a group of five ewes were underfed and mineral oil supplemented (UFMO).

[2]The non-contaminated concentrate was made of 50% dehydrated beet pulp and 50% corn grain and the contaminated concentrate was made of 97.6% non-contaminated concentrate and 2.4% contaminated rapeseed oil (fresh matter basis).

amounts were weekly adjusted depending on ewe BW to cover the expected level of MER (0.23 MJ net energy.kg BW$^{-0.75}$; [13]). Control and UFMO ewes received daily 15 or 25 g of minerals and vitamin premix (Phosphore/Calcium = 18/5; Chauveau, Cholet, France), respectively, and had free access to drinking water.

## Sampling, measurements, and chemical analyses

**Feed.** Feedstuffs offered to animals were individually and daily weighed as well as potential refusals (only straw). In order to exclude uncontrolled straw ingestion, ewes were housed on metal shelves. Subsamples of feedstuffs and refusals were daily collected and weekly pooled for DM determination (60˚C, 48h). Feedstuff subsamples were weekly collected, pooled for all the duration of the experiment (hay, straw, contaminated concentrate and non-contaminated concentrate) and ground through a 1-mm sieve prior to the analyses of DM, ash, crude protein, non-detergent and acid-detergent fibers [14], total lipids [15], *in vitro* pepsin-cellulase digestibility, and for non-contaminated concentrate, *in vitro* 1-h nitrogen degradability [16]. Energy and proteins truly digestible in the small intestine contents were estimated for each feedstuff from ash, crude protein, acid-detergent fiber, *in vitro* pepsin-cellulase digestibility, ether extract and *in vitro* 1-h nitrogen degradability (only for non-contaminated concentrate for this last two parameters) using the INRA PrevAlim software [17]. The concentrations of TCDD and PCBs were also analysed in all feedstuff. Feedstuff chemical composition is presented in S1 Table, whereas feedstuff, nutrients and POPs intakes are presented in Table 2.

**Body measurements.** Ewes were weekly weighed at 0800 h before feed distribution and BCS was estimated on a 0–5 scale [18] at days -1, +6, +20, +34 and +57 according to the start of the depuration period. Energy and protein balances were weekly and individually calculated based on BW and estimated energy and protein intakes [13].

**Blood.** Blood samples were collected by venipuncture from the jugular vein on days 0, +7, +21, +35 and +57 of the depuration period at 0800 h before feed distribution. Some samples were collected on lithium-heparin tubes and kept on ice for less than 1h before plasma was separated by centrifugation (1 400$g$, 15min, 4˚C), and frozen at -20˚C before analyses of metabolites. Other samples were collected in tubes with clot activator (SiO$_2$), maintained at 4˚C during 20h, and then serum was separated by centrifugation (1 400$g$, 25min at 20˚C) and one subsample was kept at -80˚C for total lipid analysis, while the other one was kept at -20˚C for TCDD and PCBs analyses. Plasma NEFA, glucose, BOH and serum total lipid concentrations were analysed spectrophotometrically using commercial kits (Wako NEFA-HR2, Oxoid Thermo-Fisher, Dardilly, France; Glucose GOD-POD, Thermo-Fisher, Vantaa, Finland; β-hydroxybutyrate, Thermo-Fisher, Vantaa, Finland;HB018, Cypress Diagnostics, Langdorp, Belgium, respectively) as described by Lerch et al. [4, 14]. Intra-assay CV were 0.8, 5.2, 2.5 and 3.0% for NEFA, glucose, BOH and total lipids, respectively.

**Adipose tissue.** Pericaudal subcutaneous AT (PSAT) was harvested by biopsy at 0800 h on days -1, +7, +22 and +35 of the depuration period. On day +57, PSAT was exhaustively collected and weighed, and then after the ewes were slaughtered. The PSAT was biopsied 5–10 cm above the tail head insertion, alternatively on the right and left sides. Skin was incised 5 cm-long with a scalpel under local subcutaneous anaesthesia (5 mL of 2% lidocaine; Lurocaïne, Vétoquinol, Lure, France). The cut-off was then sutured and treated with antiseptic spray (chlorhexidine, Cicajet 18, Virbac, Carros, France). The adipocyte volume determination was performed on a subsample of around 50 mg PSAT rapidly excised and placed immediately in physiological saline at 39˚C and then fixed with osmium oxide tetroxide during at least one week [19]. The arithmetic means of diameter and volume of 350–450 fixed adipocytes with diameter $\geq$ 25 μm were thereafter determined under microscope. Vessels and connective

**Table 2. Ingredients, nutrients and POPs intakes of ewes[1] during the buffering and depuration periods.**

| Item | Treatment | Buffering period (mean ± SD) | Depuration period | | | | | SEM | P-value | | |
|---|---|---|---|---|---|---|---|---|---|---|---|
| | | | Day | | | | | | T[2] | Day | T×Day |
| | | | 7 | 21 | 35 | 56 | | | | | |
| Ingredient intakes (g of DM.day$^{-1}$) | | | | | | | | | | | |
| Straw | CTL | 217±107 | 185 | 229 | 254 | 265 | | 23 | 0.84 | 0.32 | 0.60 |
| | UFMO | 278±63 | 251 | 245 | 240 | 236 | | | | | |
| Hay | CTL | 309±19 | 319 | 320 | 307 † | 304 * | | 12 | 0.08 | <0.001 | <0.01 |
| | UFMO | 343±10 | 309$^a$ | 287$^b$ | 269$^c$ | 259$^c$ | | | | | |
| Non-contaminated concentrate | CTL | 397±23 | 427 | 411 | 418 | 409 | | 11 | <0.001 | <0.001 | <0.001 |
| | UFMO | 441±13 | 46 | - | - | - | | | | | |
| Mineral oil | CTL | - | - | - | - | - | | 2 | <0.001 | <0.001 | <0.001 |
| | UFMO | - | 50 | 60 | 57 | 54 | | | | | |
| Total DM | CTL | 923±138 | 931 ** | 960 ** | 979 ** | 978 ** | | 41 | <0.001 | 0.30 | 0.04 |
| | UFMO | 1064±67 | 657$^a$ | 592$^b$ | 566$^b$ | 549$^b$ | | | | | |
| POPs intakes[3] | | | | | | | | | | | |
| PCB 126 (ng.day$^{-1}$) | CTL | 0.44±0.09 | 0.42 | 0.45 † | 0.47 * | 0.47 ** | | 0.03 | 0.04 | 0.48 | 0.07 |
| | UFMO | 0.51±0.05 | 0.39$^a$ | 0.37$^{ab}$ | 0.35$^b$ | 0.34$^b$ | | | | | |
| PCB 153 (µg.day$^{-1}$) | CTL | 0.12±0.02 | 0.12 * | 0.12 ** | 0.12 ** | 0.12 ** | | 0.01 | <0.01 | 0.25 | 0.03 |
| | UFMO | 0.13±0.01 | 0.09$^b$ | 0.08$^a$ | 0.08$^a$ | 0.08$^a$ | | | | | |
| Nutrient intakes (g.day$^{-1}$) | | | | | | | | | | | |
| Neutral detergent fiber | CTL | 488±96 | 479 | 509 * | 522 ** | 526 ** | | 28 | 0.01 | 0.48 | 0.07 |
| | UFMO | 571±51 | 413$^a$ | 380$^{ab}$ | 364$^b$ | 354$^b$ | | | | | |
| Acid detergent fiber | CTL | 283±61 | 275 | 296 * | 304 * | 307 ** | | 17 | 0.04 | 0.48 | 0.08 |
| | UFMO | 333±33 | 255$^a$ | 237$^{ab}$ | 227$^{ab}$ | 221$^b$ | | | | | |
| Energy (MJ net energy.day$^{-1}$) | CTL | 4,7±0.5 | 4.9 * | 4.9 * | 5.0 * | 4.9 * | | 0.2 | <0.001 | 0.02 | <0.01 |
| | UFMO | 5,4±0.2 | 2.2$^a$ | 1.8$^b$ | 1.7$^b$ | 1.7$^b$ | | | | | |
| PDI[4] | CTL | 39±4 | 41 * | 41 * | 41 * | 41 * | | 1 | <0.001 | <0.001 | <0.001 |
| | UFMO | 44±2 | 20$^a$ | 16$^b$ | 15$^b$ | 15$^b$ | | | | | |
| Fat (ether extract)[5] | CTL | 14±1 | 14.0 ** | 14.3 ** | 14.0 ** | 13.8 ** | | 1.8 | <0.001 | <0.001 | <0.001 |
| | UFMO | 15±1 | 57.4$^d$ | 66.4$^a$ | 63.0$^b$ | 60.0$^c$ | | | | | |
| Fat (lipids)[6] | CTL | 15±2 | 16.0 ** | 16.1 ** | 16.3 ** | 16.1 ** | | 1.9 | <0.001 | <0.001 | <0.001 |
| | UFMO | 18±1 | 58.3$^d$ | 67.0$^a$ | 63.7$^b$ | 60.7 c | | | | | |
| PDI[4] Balance (% MR[7]) | CTL | 84±3 | 74 * | 76 * | 77 * | 77 * | | 1 | <0.001 | 0.19 | <0.001 |
| | UFMO | 88±4 | 35$^a$ | 30$^b$ | 31$^b$ | 30$^b$ | | | | | |
| Energy balance (% MR[7]) | CTL | 89±6 | 93 * | 96 * | 99 * | 98 * | | 2 | <0.001 | 0.42 | <0.001 |
| | UFMO | 101±7 | 42$^a$ | 36$^b$ | 36$^b$ | 37$^b$ | | | | | |

[1]Four ewes received a control well-fed and non-supplemented treatment (CTL), while five ewes received an underfed and mineral oil supplemented treatment (UFMO).

[2]T: Treatment.

[3]TCDD was not detected in straw, hay and non-contaminated concentrate (< Limit of detection).

[4]PDI: Proteins truly digestible in the small intestine.

[5]Ether extract after acid hydrolysis according to the method 920–39 (AOAC, 1997).

[6]Cold extraction according to Folch et al., (1957).

[7]MR: Maintenance requirements.

$^{a-d}$Means within a row and between days with different letters differ at $P \leq 0.05$.

†, *, **Means within a column and between treatments tend to differ at $P \leq 0.10$ (†), differ at $P \leq 0.05$(*) or differ at $P \leq 0.01$ (**).

tissue were suppressed from another 2.5–5 g subsample of PSAT and then this sample was frozen at -20˚C, lyophilized and finely ground prior to TCDD and PCBs analysis.

**Faeces.**   On days 0, +7, +22, +35 and +55 of the depuration period (at around 0800 h), 50–150 g of faeces were individually collected straight from rectum. Faecal samples were lyophilized and pooled based on equal amounts of dry faeces from each ewe by treatment (CTL or UFMO) and by date (0, +7, +22, +35, +55 d, $n$ = 10). Pools were ground through a 1-mm sieve before total lipids [15], TCDD and PCBs analyses.

**TCDD, PCBs 126 and 153 analyses.**   Concentrations of TCDD, PCBs 126 and 153 were determined according to ISO/IEC 17025:2005 fully accredited methods (except for faeces), which have been slightly adapted from previously described methods [20, 21]. Methods are detailed in S2 File.

## Calculations and statistical analyses

As proposed by Atti and Bocquier [22], an allometric equation was adjusted between the neperian logarithms of BW and PSAT total lipid weight determined at slaughter (day +57):

$$\text{Log (PSAT lipids weight, g)} = 4.74 \text{ (SE 1.42)} \times \log \text{(BW, kg)} - 15.11 \text{ (SE 5.68)}; \ P = 0.01, \ rSD = 0.58, \ rCB = 15.2\%, \ R^2 = 0.614, n = 9.$$

From this equation, PSAT total lipid weight was estimated from BW on days -1, +6, +20, +34 of the depuration period. By multiplying those weights by respective PSAT TCDD and PCBs 126 and 153 concentrations, PSAT POPs burdens were individually estimated for those four dates, while they were directly determined by exhaustive sampling and analysis for day +57.

Data collected along the depuration period were analysed by ANOVA for repeated measures using the MIXED procedure of SAS (SAS Institute Inc., Cary, NC, 2003). The model included a covariate term (i.e. individual data obtained at day -1 of the depuration period), the depuration treatment (CTL and UFMO), the day and the interaction treatment × day as fixed effects, and the ewe as random effect. An autoregressive first order covariance structure was used for feedstuff/nutrient intakes and energy/protein balances, while a spatial power covariance structure was used for all the other parameters (when time intervals between measurements were unequal). Logarithmic transformation of experimental data was performed for plasma NEFA and BOH to comply with the assumptions of normality and homoscedasticity of residuals. In this case, least squares means and SEM were calculated from untransformed values, whereas declared $P$-values reflect statistical analysis of transformed data. When treatment × date interaction was significant, treatment differences at each date were determined based on the results of the slice option of SAS. Significance was declared at $P \leq 0.05$, and trends were considered at $0.05 < P \leq 0.10$. Values reported are least square means and standard error of the mean. Codes for statistical analyses are provided in S3 File.

## Results

### Nutritional and physiological status

By design, during the depuration period energy balance of UFMO ewes was in average -3.1 MJ.day$^{-1}$, representing 37% of MER. In contrast, the energy balance of CTL ewes was -0.2 MJ. day$^{-1}$, covering 97% of MER (Table 2). Between days 7 and 57 of the depuration period, BW, BCS and adipocyte diameter in CTL ewes remained stable ($P > 0.10$), whereas they significantly ($P < 0.05$) decreased by 11.5 kg, 1.0 point, and 10 μm, respectively, in UFMO ewes (Table 3). Plasma BOH concentration was higher ($P < 0.05$) in UFMO than in CTL ewes from days +7 to +57. Time and treatment had no effect ($P > 0.10$) on plasma NEFA and glucose and on serum lipids concentrations all over the depuration period (Table 3).

**Table 3. Intakes, body measurements and plasma metabolites concentrations of ewes groups[1].**

| Item | Treatment | Buffering period (mean±SD) | Depuration period | | | | | | P-value | | |
|---|---|---|---|---|---|---|---|---|---|---|---|
| | | | Day | | | | | | | | |
| | | | 7 | 21 | 35 | 57 | SEM | T[2] | Day | T×Day |
| Body measurement | | | | | | | | | | | |
| BW (kg) | CTL | 62.0±6.3 | 61.6[ab] | 61.9[a] * | 60.1[b] ** | 60.6[ab] ** | 1.2 | 0.01 | <0.001 | <0.001 |
| | UFMO | 64.2±5.8 | 61.1[a] | 57.5[b] | 53.6[c] | 49.6[d] | | | | |
| Body condition score (0–5) | CTL | 3.6±0.3 | 3.7[ab] | 3.6[b] | 4.0[a] * | 3.5[b] * | 0.1 | 0.05 | <0.001 | 0.01 |
| | UFMO | 3.6±0.4 | 3.6[a] | 3.6[a] | 3.4[a] | 2.6[b] | | | | |
| Adipocyte diameter (µm) | CTL | 78±11 | 77[ab] | 71[b] | 79[a] | 76[ab] | 2 | 0.85 | 0.02 | 0.36 |
| | UFMO | 83±10 | 80[a] | 74[b] | 75[ab] | 70[b] | | | | |
| Plasma metabolites | | | | | | | | | | | |
| Total lipids (mg.dL$^{-1}$) | CTL | 238±26 | 218 | 209 | 238 | 229 | 12 | 0.19 | 0.29 | 0.64 |
| | UFMO | 193±25 | 175 | 199 | 197 | 209 | | | | |
| NEFA[3] (µM) | CTL | 730±383 | 580 | 670 | 531 | 598 | 106 | 0.12 | 0.24 | 0.91 |
| | UFMO | 675±227 | 795 | 959 | 761 | 873 | | | | |
| BOH[4] (µM) | CTL | 365±82 | 263 * | 297 ** | 269 * | 315 * | 46 | 0.02 | 0.06 | 0.85 |
| | UFMO | 257±29 | 471 | 553 | 467 | 555 | | | | |
| Plasma glucose (mg.dL$^{-1}$) | CTL | 70±10 | 71 | 75 | 66 | 67 | 2 | 0.76 | 0.52 | 0.36 |
| | UFMO | 67±9 | 70 | 71 | 73 | 69 | | | | |

[1]Four ewes received a control well-fed and non-supplemented treatment (CTL), while five ewes received an underfed and mineral oil supplemented treatment (UFMO).

[2]T: Treatment.

[3]NEFA: Non-esterified fatty acids.

[4]BOH: Beta-hydroxybutyrate.

[a-d]Means within a row and between days with different letters differ at P≤0.05.

[*,**]Means within a column and between treatments differ at P≤0.05(*) or P≤0.01 (**).

## POPs kinetics within ewe organism

**Oral exposure.** During the exposure period, ewes ingested daily 280±35 pg TCDD, 285 ±35 pg PCB 126 and 281±35 ng PCB 153.kg BW$^{-1}$. During the buffering and depuration periods, the ingestion was divided on average by 40 and 150 for PCB 126 and PCB 153, respectively, while it was negligible for TCDD (i.e. lower than the limit of detection in hay, straw and non-contaminated concentrate, Table 2 and S1 Table).

**Faecal excretion.** As the results represent pooled data, no statistical analysis could be performed on faecal data. Besides, the results could be open to criticism as they were occasionally obtained (i.e. one rectal spot sample at 5 fixed days before the start or during the depuration period). However, the effects of the treatments at those 5 dates were in full accordance with those obtained from faecal pool samples composited for each ewe along the 8-week depuration period based on 16 single samples (twice weekly, every 3 to 4 days) which allowed to determine finely the masses of POP excreted through the faeces along this period of time. Those results are presented in the second companion paper [23]. Faecal lipid concentrations of CTL ewes remained between 2.1 and 2.4% (on DM basis) along the depuration period, while faecal lipid concentrations of UFMO ewes increased sharply from 2.1 to 12.4% between days -1 to +7, and then remained high (S3 Table). Whatever the mode of expression of faecal concentrations of POPs (per g of lipids or per g of DM), they remained stable in CTL ewes along the depuration period (from day -1 to day +35) but they raised suddenly at day +55 for PCBs 126 and 153. Conversely, the faecal POPs concentrations expressed on DM numerically increased by 2.0-fold for TCCD and PCB 126 and by 2.6-fold for PCB 153 from days -1 to +55 in UFMO

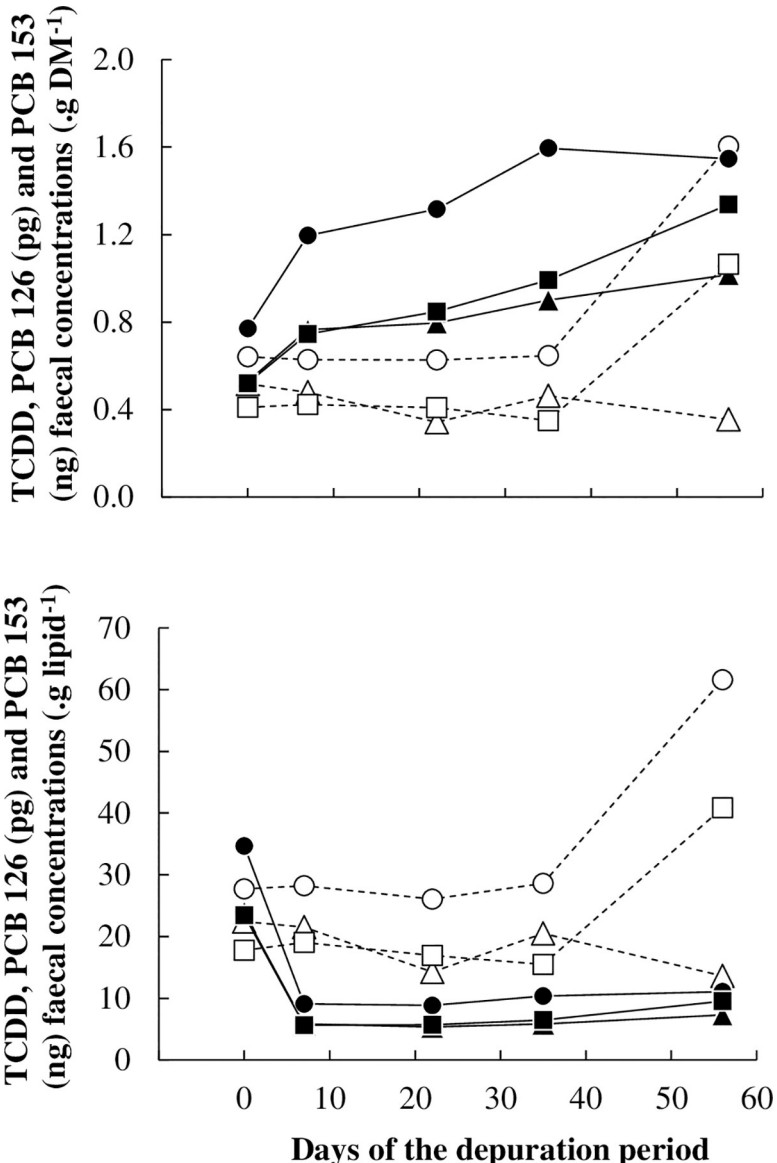

**Fig 1. Time pattern kinetics of pollutants faecal concentrations based on dry matter (DM) or on lipids; with TCDD (▲, Δ), PCB 126 (●, ○) and PCB 153 (■, □) in ewes receiving a control well-fed and non-supplemented treatment (CTL: Δ, ○, □) or an underfed and mineral oil supplemented treatment (UFMO: ▲, ●, ■).** Each point represents individual results obtained from pools of faeces by treatment and by date.

ewes. On the contrary, when expressed on lipids, faecal POPs concentrations were 4.0-fold decreased from days -1 to +7 in UFMO ewes, and then remained low (Fig 1 and S3 Table).

**Pollutant kinetics in serum and adipose tissue.** During the depuration period, serum TCDD and PCB 153 concentrations remained unchanged and equivalent for both treatments from days +7 to +57, whereas serum PCB 126 concentration was significantly decreased between days +7 and +57 in CTL ewes ($P < 0.05$), and slightly but not significantly decreased in UFMO ewes during the same period (Fig 2, S2 Table). Nonetheless, treatment had no effect on serum concentrations whatever the POPs and the day of the experiment (except on day +21 for PCB 126 concentration).

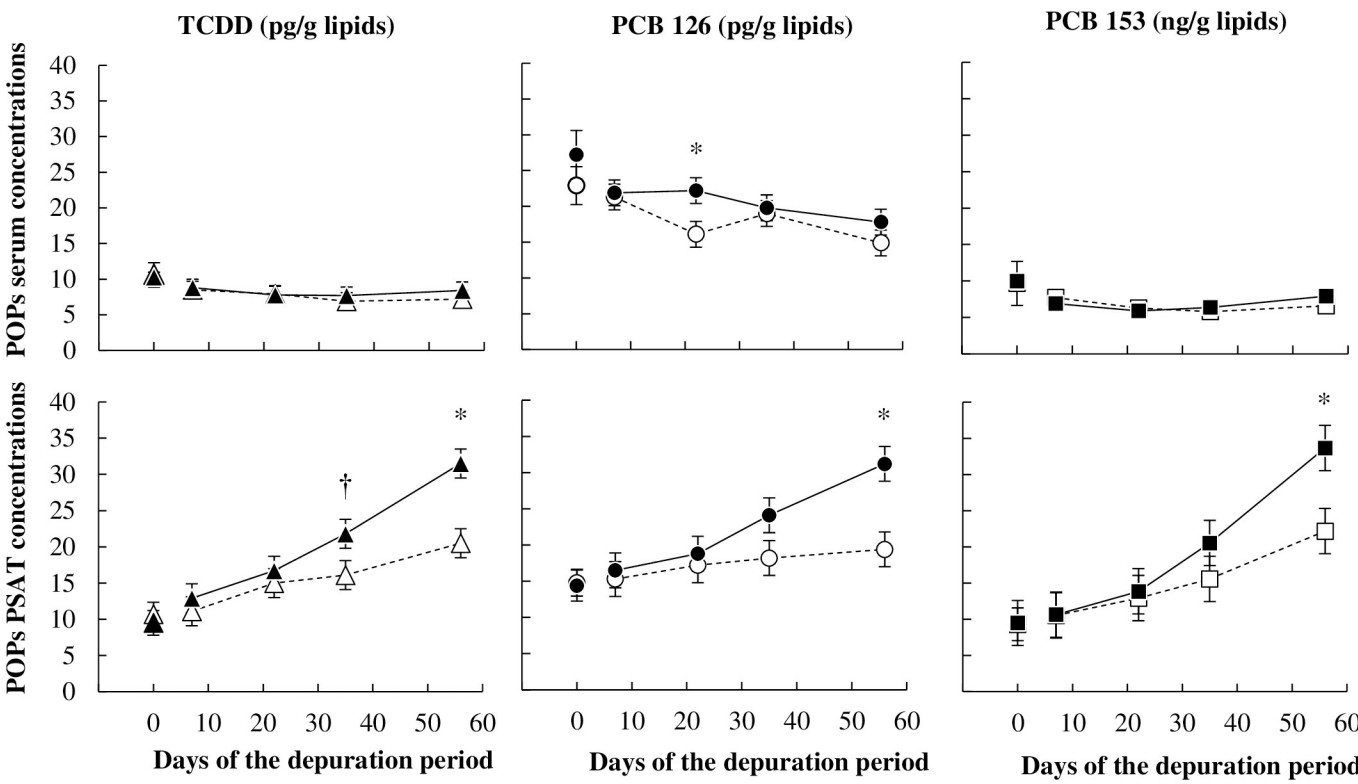

**Fig 2. Time pattern kinetics of pollutants concentrations in serum and pericaudal subcutaneous adipose tissue (PSAT) of ewes receiving a control well-fed and non-supplemented treatment (CTL: Δ, ○, □) or an underfed and mineral oil supplemented treatment (UFMO: ▲, ●, ■).** Each point represents the least-squares mean, and error bars indicate SEM except at day 0, for which they represent the mean and SE. The * symbol indicates a significant ($P \leq 0.05$) and † a tendency toward significance ($P \leq 0.10$) for treatment effect on the considered day.

Whatever the treatment, POPs concentrations in PSAT increased during the depuration period ($P < 0.01$), but they tended to be different between treatment for TCDD and PCB 126. Increases were higher in UFMO than in CTL ewes, as POPs concentrations in PSAT after 57 days of depuration were 1.5 to 1.6-fold higher in UFMO than in CTL ewes ($P < 0.05$, Fig 2, S2 Table). Throughout the depuration period, estimated PSAT lipid weight remained unchanged in CTL ewes, whereas it decreased linearly and significantly ($P < 0.05$) by 2.7-fold until 33 g of lipids between days +7 and +57 in UFMO ewes (Fig 3, S2 Table). Therefore, PSAT lipid weight was lower in UFMO than in CTL ewes from day +35 to day +57 ($P < 0.05$). Estimated PSAT TCDD and PCB 153 burdens increased at day +57 in CTL ($P < 0.05$ compared to day -1), but remained unchanged ($P > 0.10$) in UFMO ewes. Conversely, estimated PSAT PCB 126 burden remained unchanged in CTL ewes while it dropped ($P < 0.05$) in UFMO ewes. Consequently at the end of the depuration period, PSAT TCDD, PCBs 126 and 153 estimated burdens were 1.6-fold lower ($P < 0.05$) in UFMO than in CTL ewes (Fig 4, S2 Table).

## Discussion

The novelty of the present study is to report together POPs kinetic and mass balance data along with quantification of lipids dynamics at the whole ruminant body scale, with two dietary treatments inducing contrasted changes in AT mobilisation and faecal lipid excretion rates. Such a combined toxicokinetics and animal physiology approach is scarce in literature, unless it is a key step in order to understand the mechanisms which link lipophilic POPs and lipid fluxes in the ruminant organism.

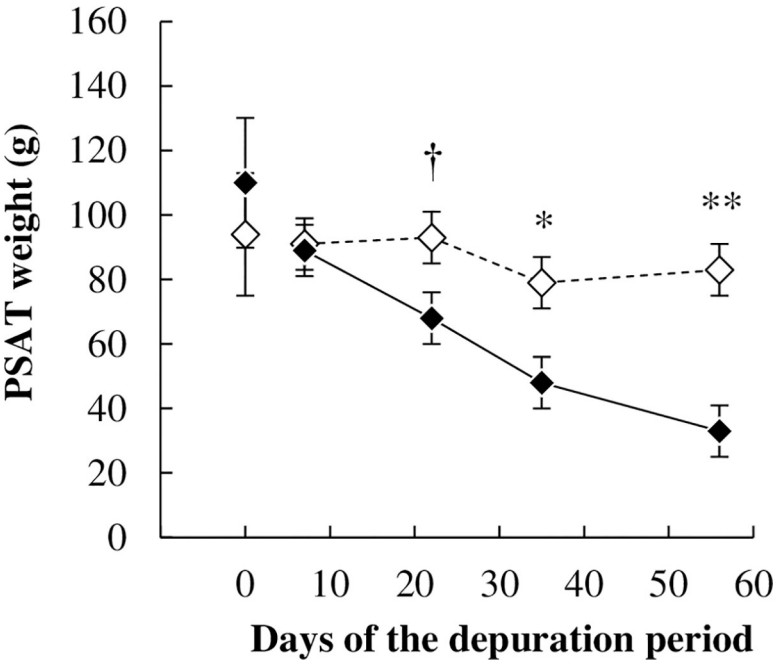

**Fig 3. Time pattern kinetics of pericaudal subcutaneous adipose tissue (PSAT) weight in ewes receiving a control well-fed and non-supplemented treatment (CTL: ◇) or an underfed and mineral oil supplemented treatment (UFMO: ◆).** Each point represents the least-squares mean, and error bars indicate SEM except at day 0, for which they represent the mean and SE. The * and ** symbols indicate a significant ($P \leq 0.05$ and $P \leq 0.01$, respectively) and † a tendency toward significance ($P \leq 0.10$) for treatment effect on the considered day.

## Blood metabolites and body fatness

Plasma BOH and total lipid concentrations in ewes fed close to 100% of MER comply with literature [4, 24]. In contrast, NEFA concentration was surprisingly high (637±315 µM, mean ± SD of buffering and depuration periods values for CTL ewes) when compared to values obtained in non-lactating well-fed ewes (60–350 µM; [4, 25, 26]). This discrepancy may originate from the 17-hour interval between the last sub-*ad libitum* diet feeding and blood

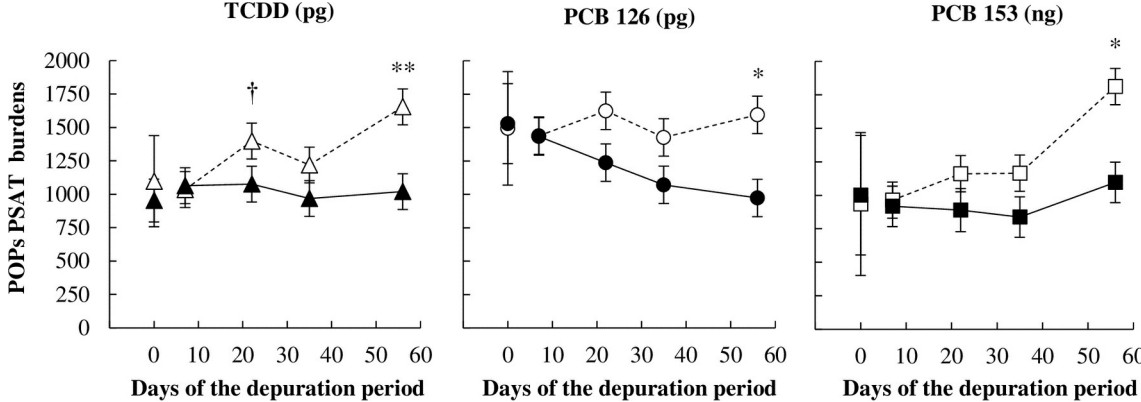

**Fig 4. Time pattern kinetics of TCDD, PCB 126 and 153 pericaudal subcutaneous adipose tissue (PSAT) burdens in ewes receiving a control well-fed and non-supplemented treatment (CTL: Δ, ○, □) or an underfed and mineral oil supplemented treatment (UFMO: ▲, ●, ■).** Each point represents the least-squares mean, and error bars indicate SEM except at day 0, for which they represent the mean and SE. The * and ** symbols indicate a significant ($P \leq 0.05$ and $P \leq 0.01$, respectively) and † a tendency toward significance ($P \leq 0.10$) for treatment effect on the considered day.

sampling. Higher plasma BOH concentration and a trend to higher plasma NEFA concentration in UFMO compared to CTL ewes from day +7 ascertain that they adapted their metabolism in response to undernutrition by mobilizing their body lipid reserves. This mobilization after 57 days at 37% of MER elicited losses of -1.8 kg BW, -0.13 point BCS and -6 μm of adipocytes diameter per week. Those decreases were higher than in non-lactating ewes fed at 35–40% of MER during 28 days (-1.3 kg BW, -0.08 point BCS per week and -3.5 μm of adipocytes diameter per week; [4]). A putative explanation of this discrepancy is that MO supplementation may have reduced the diet digestibility, leading to overestimation of the calculated energy balance of UFMO ewes in the present study.

### Effects of undernutrition combined with mineral oil supplementation on TCDD and PCBs kinetics of depuration

**Faecal excretion of POPs.** According to several authors [27, 28], lipophilic and non-polar POPs excreted as parent compounds from body compartments to faeces originate from passive diffusion across the intestinal wall along the concentration gradient of the pollutants between blood fat and digestive contents fat, rather than from biliary excretion or exfoliation of intestinal enterocytes. This phenomenon probably occurred in the current study, leading to an enrichment of 2.0 to 2.6-fold in faecal POPs concentrations (on DM basis), which were concomitant with the increase in faecal lipid concentration by 6-fold due to UFMO treatment from days -1 to +55. Similarly, supplementing diets with 50 g MO per day elicited enhancements in faecal POPs concentrations of 2 to 3-fold magnitude in dairy goats contaminated by mirex [6] and in growing lambs contaminated by hexachlorobenzene [5]. However, contrary to the current study, these animals fed diets covering their energy requirements. Thus, based on this indirect comparison between the results of the present study and those of Rozman et al. [5, 6], undernutrition does not seem to improve the increase in faecal POPs concentrations linked to MO supplementation. As faeces/serum POPs concentration ratios (on lipid basis) were decreased from 2.0, 1.4 and 2.5 in CTL ewes to 0.7, 0.5 and 1.0 in UFMO ewes for TCDD, PCBs 126 and 153, respectively, it seems that the equilibrium of POPs concentrations between serum and faeces was not reached in UFMO ewes (ratio faeces/serum < 1) in contrary to CTL ewes (ratio faeces/serum > 1). This suggests that the POPs transfer rate from blood to the intestine lumen was the limiting step of the enhancement of POPs faecal excretion due to UFMO. It putatively may have two origins:

i) The inability of highly lipophilic molecules such as TCDD, PCBs 126 and 153 (log $K_{ow}$ > 6.8; [29], with $K_{ow}$ the octanol/water partition coefficient) to easily cross through the wall of the intestinal tract by passive diffusion, as previously reported in humans [30]. In mice, undernutrition (50% of *ad libitum* level) combined with 10% olestra (i.e. another non-absorbable lipid) in total diet during 15 days increased by more than 30-fold the faecal excretion of hexachlorobenzene, whereas hexachlorobenzene faeces/plasma concentration ratio remained stable (estimation of ≈ 0.5) compared to *ad libitum* or underfed and non-supplemented diets [31]. Such discrepancies among studies could originate, at least in part, from differences in terms of POPs molecules properties (lower lipophilicity of hexachlorobenzene: log $K_{ow}$ of 5.5, than that of TCDD, PCBs 126 and 153: log $K_{ow}$ > 6.8; [29]).

ii) The limited available amount of POPs in blood circulating in the splanchnic area, when compared to the total POPs body burden. Concerning this last assumption, undernutrition in ruminants is known to affect more severely the weight of digestive tract than the total BW [32, 33]. It also decreases the blood total volume and perfusion in the splanchnic area [33, 34]. Subsequently, the available exchange surface for POPs diffusion between blood and intestinal lumen is expected to be reduced.

**Kinetics of POPs in serum.** In a previous study with non-lactating ewes, we observed a 7-day initial drop in serum PCBs 138, 153, and 180 concentrations after oral exposure had ceased [4]. This was attributed to the buffering effect of rumen, which delays the absorption and of dietary constituents, including POPs. Therefore, we implemented an 8-day buffering period in the current study to avoid this disturbance. In spite of this precaution, serum POPs concentrations tended to drop from days -1 to +7 of the depuration period whatever the treatment. Thus, it seems that the end of absorption and initial distribution were still observable between 8 to 16 days once the exposure had ceased (i.e. days -1 to +7 of depuration period).

From days +7 to +57, POPs serum concentrations tended to decrease only slightly whatever the treatment and the POP. This may be explained by the very limited metabolism [35] and faecal excretion of POPs. Overall during this period, UFMO treatment did not increase serum POPs concentrations compared to CTL, conversely to our initial hypothesis. In non-lactating ewes underfed but not supplemented with non-absorbable lipids, serum PCB 138, 153 and 180 concentrations were increased concomitantly to body fat mobilization [4]. Thus, it seems that supplementing ewes with MO counteracted the expected increases in blood POPs concentrations due to release from AT elicited by undernutrition by increasing blood depuration through faecal excretion.

**Kinetics of POPs in adipose tissue.** In CTL ewes, PSAT concentrations of TCDD and PCB 153 increased by 1.9 to 2.1-fold from days +7 to +57. Such increases could not be explained by a concentration effect, since the volume of the diffusion pool (i.e. PSAT lipid weight) remained unchanged. We suspect that it arose from a late redistribution of POPs toward the subcutaneous AT [11, 36]. Indeed, following absorption, POPs rapidly distribute from lymph and blood to the highly-perfused tissues within a few days. Thereafter, a second-step redistribution sometimes occurs toward the slowly-perfused tissues (such as AT). This phenomenon might be slow and last for several weeks, especially when slowly-perfused AT compartment volume is large, when POPs elimination rate is low and/or when POPs amount available for exchange in the blood distribution compartment is reduced (i.e. when serum/AT concentrations ratio is low). Such kind of late redistribution pattern toward AT was well described during a 388-day depuration study involving rhesus monkey dosed with mirex, a highly lipophilic (log $K_{ow}$ of 6.9; [29]) and poorly metabolized POP, showing a ratio serum/AT on lipid basis around 0.4 [36]. It was also reported in non-lactating cows exposed to TCDD after they had been submitted to 27-day depuration [11]. In CTL ewes, the delayed redistribution of TCDD and PCB 153 probably occurred (mean serum/AT concentrations ratios on lipid basis of 0.43 and 0.38, respectively), whereas the redistribution pattern of PCB 126 seemed shorter (mean serum/AT ratio on lipid basis of 0.95) and thus only marginally affected PSAT concentration along the depuration period. The consequences were that over the 57-day depuration period the estimated PSAT burden of TCDD and PCB 153 in CTL ewes increased by 1.5 and 1.9-fold, respectively, whereas the one of PCB 126 remained unchanged.

In UFMO ewes, TCDD and PCB 153 concentrations in PSAT also encountered the highest increase along the depuration period compared to PCB 126 (3.3 and 3.5-fold, respectively *vs.* 2.2-fold for PCB 126). Nonetheless, in contrast to CTL ewes, a putative redistribution pattern was less easy to be perceived, due to confounding processes affecting POPs distribution and excretion linked to both undernutrition and MO supplementation. Indeed, on one hand, UFMO induced a 2.7-fold decrease in PSAT volume (i.e. estimated PSAT lipid weight) between days +7 and +57, while, on the other hand, it increased the faecal excretion of POPs. These combined processes achieved a decrease in the estimated PSAT PCB 126 burden by 1.5-fold, whereas no effect was observed for TCDD and PCB 153. Thus, our data suggest that when UFMO ewes mobilized lipids, they did not eliminate POPs from AT in proportional amounts. Mice receiving diets supplemented with 10% unabsorbable lipids (i.e. olestra) at 50%

of their *ad libitum* intake lost similar amounts of epididymal AT weight (2.1-fold) and hexa-chlorobenzene burden (1.6-fold) after 15 days of depuration [31]. Under similar dietary conditions, an even higher loss in body PCBs burden (3.1-fold) compared to body lipids weight (1.5-fold) was achieved in growing chickens dosed with Aroclor 1254 after 21-day depuration [8]. Differences in POPs molecules properties, animal models, experimental design and/or nutrition levels (i.e. undernutrition intensity) could explain such discrepancies between studies.

## Conclusions

The UFMO treatment increased concentrations of TCDD and PCBs in faeces compared to CTL. However, the 2-fold magnitude increase did not exceed that previously observed in well-fed MO supplemented small ruminants for hexachlorobenzene or mirex. After 57 days of depuration, 1.6-fold lower PSAT TCDD and PCBs estimated burdens were observed for UFMO when compared to CTL ewes. This was putatively, and at least in part, linked to the enhancement of POPs faecal excretion. The second companion paper [23] focuses on POPs mass balance and distribution in order to determine whether a similar decrease occurred at the whole body scale, as well as in other internal AT depots (i.e. perirenal, mesenteric, intramuscular. . .), which are known to be differentially perfused by blood and sensitive to mobilization in response to undernutrition. Together, the results of the current study will be helpful for fine-tune effective on-farm feeding strategies for depurating ruminants from POPs (dioxins and PCBs), which will allow to alleviate economic and social damages associated with disposal of contaminated herds in case of food safety crisis.

## Supporting information

**S1 Table. Chemical composition, nutritional values, and dioxin (TCDD) and polychlori-nated biphenyls (PCBs) concentrations of feedstuffs.**
(DOCX)

**S2 Table. Dioxin (TCDD) and polychlorinated biphenyls (PCBs) concentrations and burdens in blood serum and pericaudal subcutaneous adipose tissue of exposed ewes.**
(DOCX)

**S3 Table. Faecal concentrations and daily flows of lipids and POPs (in dry matter and lipids basis) of exposed ewes.**
(DOCX)

**S1 File. Contaminated rapeseed oil and mineral oil description.**
(DOCX)

**S2 File. POPs analyses method.**
(DOCX)

**S3 File. Codes for statistical analyses.**
(DOCX)

## Acknowledgments

The authors thank B. Mallet, A. Guittard, Y. Thomas, P. Payard and D. Roux (INRA, UE 1414) for feeding and management of ewes; N. Besné (INRA, UE 1295 PEAT, Nouzilly, France) for the concentrate manufacture; S. Collange, J. Mongiat (INRA, UE 1414), D. Durand and I. Constant (Université Clermont Auvergne, INRA, VetAgro Sup, UMR 1213) for biopsies and

adipocytes cellularity measurements; and P. Hartmeyer and C. Grandclaudon (Université de Lorraine, INRA, UR AFPA) for technical support along the study.

## Author Contributions

**Conceptualization:** Ronan Cariou, Sylvain Lerch.

**Data curation:** Lucille Rey-Cadilhac, Sylvain Lerch.

**Formal analysis:** Lucille Rey-Cadilhac, Sylvain Lerch.

**Funding acquisition:** Ronan Cariou, Anne Ferlay, Sylvain Lerch.

**Investigation:** Lucille Rey-Cadilhac, Carole Delavaud, Sébastien Alcouffe, Philippe Marchand.

**Methodology:** Ronan Cariou, Catherine Jondreville, Carole Delavaud, Yannick Faulconnier, Stefan Jurjanz.

**Project administration:** Lucille Rey-Cadilhac, Anne Ferlay, Sylvain Lerch.

**Resources:** Pascal Faure, Philippe Marchand, Bruno Le Bizec.

**Software:** Lucille Rey-Cadilhac, Sylvain Lerch.

**Supervision:** Anne Ferlay, Sylvain Lerch.

**Validation:** Ronan Cariou, Anne Ferlay, Sylvain Lerch.

**Visualization:** Lucille Rey-Cadilhac.

**Writing – original draft:** Lucille Rey-Cadilhac, Anne Ferlay, Sylvain Lerch.

**Writing – review & editing:** Ronan Cariou, Anne Ferlay, Catherine Jondreville, Carole Delavaud, Yannick Faulconnier, Stefan Jurjanz.

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
