## [Decision Letter · Decision Letter 0]

20 Dec 2019

PONE-D-19-30221

Undernutrition combined with dietary mineral oil hastens depuration of stored dioxin and polychlorinated biphenyls in ewes. 1. Kinetics in blood, adipose tissue and faeces

PLOS ONE

Dear Dr. Lerch,

Thank you for submitting your manuscript to PLOS ONE. After careful consideration, we feel that it has merit but does not fully meet PLOS ONE’s publication criteria as it currently stands. Therefore, we invite you to submit a revised version of the manuscript that addresses the points raised during the review process.

lmanager.com/pone/ and select the 'Submissions Needing Revision' folder to locate your manuscript file.

To enhance the reproducibility of your results, we recommend that if applicable you deposit your laboratory protocols in protocols.io, where a protocol can be assigned its own identifier (DOI) such that it can be cited independently in the future. For instructions see: http://journals.plos.org/plosone/s/submission-guidelines#loc-laboratory-protocols

We look forward to receiving your revised manuscript.

Kind regards,

Juan J Loor

Academic Editor

PLOS ONE

Journal Requirements:

2. The link provided In your Data Availability Statement ;https://doi.org/10.15454/Z6UML7,  states that the DOI is not found.

PLOS defines a study's minimal data set as the underlying data used to reach the conclusions drawn in the manuscript and any additional data required to replicate the reported study findings in their entirety. All PLOS journals require that the minimal data set be made fully available. For more information about our data policy, please see http://journals.plos.org/plosone/s/data-availability.

Reviewers' comments:

Reviewer's Responses to Questions

**Comments to the Author**

1. Is the manuscript technically sound, and do the data support the conclusions?

Reviewer #1: Partly

2. Has the statistical analysis been performed appropriately and rigorously? 

Reviewer #1: No

3. Have the authors made all data underlying the findings in their manuscript fully available?

Reviewer #1: No

4. Is the manuscript presented in an intelligible fashion and written in standard English?

Reviewer #1: No

5. Review Comments to the Author

Reviewer #1: here.

Line 79: Here the author give the three molecules investigated in this study. However, the description above has nothing to do with these molecules. Please rewrite the introduction to focus on the dioxin and PCBs.

Line 192: Because the concentrations of TCDD and PCBs were very low in the samples, it was difficult to analyze them in small amount of samples. How much volume of blood and serum and other matrix were collected and analyzed in this study? What was the detection limit? Please provide the details.

Figure 1-4: These figures can’t be displayed in PDF version.

Please show more data in the manuscript.

6. PLOS authors have the option to publish the peer review history of their article (what does this mean?). If published, this will include your full peer review and any attached files.

Reviewer #1: No

---

## [Author Response · Author response to Decision Letter 0]

3 Feb 2020

Subject: Manuscript PONE-D-19-30221: “Undernutrition combined with dietary mineral oil hastens depuration of stored dioxin and polychlorinated biphenyls in ewes. 1. Kinetics in blood, adipose tissue and faeces“

Dear Editor,

We appreciate all of the constructive comments and thoughtful suggestions provided by the referee. All suggestions arising from the peer-review have been carefully considered during revision of the original manuscript. In all cases, the suggested changes have been carefully considered and implemented in full.

The associate changes are highlighted in red thorough the manuscript labeled 'Revised Manuscript with Track Changes’. A response to the comment from the referee is outlined in the accompanying response indicated by sentences starting with AU:. The line number reported in the response are the one of the 'Revised Manuscript with Track Changes’.

We hope that this revision would allow the manuscript to be considered acceptable for publication.

With our best regards,

L. Rey-Cadilhac and S. Lerch,

 

Journal Requirements:

AU: Our initial figures files do not meet the PLOS ONE requirements. Files were reshaped in order to meet the requirements before resubmission of this R1 version. Now, they appear and are readable in the pdf version of the manuscript at submission.

2. The link provided In your Data Availability Statement: https://doi.org/10.15454/Z6UML7, states that the DOI is not found.

AU: The https://doi.org/10.15454/Z6UML7 is the final link where the data would be available once the paper would be published (access would be opened). Until then, a private link, where the data can be found until they get public; is: https://data.inra.fr/privateurl.xhtml?token=9d8c85ad-263c-4cab-b36d-66d1a0a5c32e

PLOS defines a study's minimal data set as the underlying data used to reach the conclusions drawn in the manuscript and any additional data required to replicate the reported study findings in their entirety. All PLOS journals require that the minimal data set be made fully available. For more information about our data policy, please see http://journals.plos.org/plosone/s/data-availability.

AU: all the individual experimental data will be made publicly available on the data.inra repository, using the DOI 10.15454/Z6UML7 (available when published).

 

Reviewers Comments to the Author

Reviewer #1: here.

Line 79: Here the author give the three molecules investigated in this study. However, the description above has nothing to do with these molecules. Please rewrite the introduction to focus on the dioxin and PCBs.

AU: A sentence was added in the introduction explaining that PCBs, dioxins, and organochlorine pesticides (hexachlorobenzene and mirex) are all part of the Persistent Organic Pollutants (POPs) list defined in the Stockholm convention. See lines 47-49 and 82-85. They have close physico-chemical (especially their high lipophilicity), and toxicokinetics properties, so that the comparison can be made across POPs molecules regarding the effect of mineral supplementation or undernutrition on toxicokinetic in the introduction section. Moreover, terminology was reviewed thorough the text, retaining only “persistent organic pollutants” as a generic term in order to describe this group of molecules, and no more “lipophilic contaminants” or “lipophilic compounds”. See lines 51 and 91 in the introduction and throughout the main text.

Line 192: Because the concentrations of TCDD and PCBs were very low in the samples, it was difficult to analyze them in small amount of samples. How much volume of blood and serum and other matrix were collected and analyzed in this study? What was the detection limit? Please provide the details.

AU: This information was added to the S2 File. POPs analysis method. which described the steps, material and methods for the POPs analysis. For all matrixes, the measured concentrations were higher than the LOD/LOQ except for the TCDD in forages and non-contaminated pelleted concentrate for which the concentrations were under the LOD/LOQ.

Figure 1-4: These figures can’t be displayed in PDF version.

AU: Figures have been reformatted using the Preflight Analysis and Conversion Engine (PACE) digital diagnostic tool as advised. They are now available in the PDF version of the new manuscript.

Please show more data in the manuscript.

AU: Now the Figures 1 to 4 are readable from the main pdf manuscript. Moreover, in order to show more data in the main text, and better balance the data provided between main text and supporting information, the Table S2 about Ingredients, nutrients and POPs intakes of ewes during the buffering and the depuration periods was displaced from the Supporting information to the main text as Table 2. The content of Tables 2 and 3 was adjusted concomitantly. Moreover, the detailed individual data are available in the Data repository resources.

---

## [Decision Letter · Decision Letter 1]

5 Mar 2020

Undernutrition combined with dietary mineral oil hastens depuration of stored dioxin and polychlorinated biphenyls in ewes. 1. Kinetics in blood, adipose tissue and faeces

PONE-D-19-30221R1

Dear Dr. Lerch,

We are pleased to inform you that your manuscript has been judged scientifically suitable for publication and will be formally accepted for publication once it complies with all outstanding technical requirements.

With kind regards,

Juan J Loor

Academic Editor

PLOS ONE

Additional Editor Comments (optional):

Reviewers' comments:

Reviewer's Responses to Questions

**Comments to the Author**

1. If the authors have adequately addressed your comments raised in a previous round of review and you feel that this manuscript is now acceptable for publication, you may indicate that here to bypass the “Comments to the Author” section, enter your conflict of interest statement in the “Confidential to Editor” section, and submit your "Accept" recommendation.

Reviewer #1: All comments have been addressed

2. Is the manuscript technically sound, and do the data support the conclusions?

Reviewer #1: Yes

3. Has the statistical analysis been performed appropriately and rigorously? 

Reviewer #1: Yes

4. Have the authors made all data underlying the findings in their manuscript fully available?

Reviewer #1: (No Response)

5. Is the manuscript presented in an intelligible fashion and written in standard English?

Reviewer #1: Yes

6. Review Comments to the Author

Reviewer #1: This authors revised the manuscript according the comments of the reviwers. The article can be accepted after modification.

7. PLOS authors have the option to publish the peer review history of their article (what does this mean?). If published, this will include your full peer review and any attached files.

Reviewer #1: No